# Microplastics and Antibiotic Resistance: The Magnitude of the Problem and the Emerging Role of Hospital Wastewater

**DOI:** 10.3390/ijerph20105868

**Published:** 2023-05-18

**Authors:** Benedetta Tuvo, Michela Scarpaci, Sara Bracaloni, Enrica Esposito, Anna Laura Costa, Martina Ioppolo, Beatrice Casini

**Affiliations:** Department of Translational Research and New Technologies in Medicine and Surgery, University of Pisa, 56126 Pisa, Italy

**Keywords:** microplastic, antimicrobial resistance (AMR), antibiotic resistance genes (ARGs), hospital wastewater

## Abstract

The role of microplastics (MPs) in the spread of antibiotic resistance genes (ARGs) is increasingly attracting global research attention due to their unique ecological and environmental effects. The ubiquitous use of plastics and their release into the environment by anthropic/industrial activities are the main sources for MP contamination, especially of water bodies. Because of their physical and chemical characteristics, MPs represent an ideal substrate for microbial colonization and formation of biofilm, where horizontal gene transfer is facilitated. In addition, the widespread and often injudicious use of antibiotics in various human activities leads to their release into the environment, mainly through wastewater. For these reasons, wastewater treatment plants, in particular hospital plants, are considered hotspots for the selection of ARGs and their diffusion in the environment. As a result, the interaction of MPs with drug-resistant bacteria and ARGs make them vectors for the transport and spread of ARGs and harmful microorganisms. Microplastic-associated antimicrobial resistance is an emerging threat to the environment and consequently for human health. More studies are required to better understand the interaction of these pollutants with the environment as well as to identify effective management systems to reduce the related risk.

## 1. Introduction

Nowadays, antibiotic resistance represents a severe problem for public health, due to the increasing diffusion in the general population, animals, and the environment of Antibiotic Resistant Bacteria (ARB) with the resistance phenotype extended to several classes of antibiotics. Antimicrobial Resistance Genes (ARGs) have been detected in surface and underground water sources and sediments [1] and it is widely demonstrated that abundance and diversity of ARB in the environment are closely related to the impact of human activities.

Environmental contamination by antibiotics and their biologically active metabolites significantly contribute to worsening this condition by exercising a selection effect on microorganisms, even at subtherapeutic concentrations. For this reason, they can represent a risk to health at a concentration range as low as ng/L-µg/L.

In Italy, antimicrobial consumption is among the highest in Europe and, according to the ECDC, our country has one of the highest rates of antibiotic resistance for several pathogens [2]. 

In addition, the use of antimicrobials in veterinary medicine may worsen this situation [3], although between 2011–2018, a reduction of their use of more than 3–4% has been noted, due to the One Health approach promoted by the WHO [1,4].

Several reports have already described the binding of these resistance determinants with microplastics (MPs), considered as a “Trojan-Horse” in the absorption and release of ARB and ARGs [5]. 

In this context, the role of wastewater in spreading these contaminants is emerging, especially for hospital wastewater, which contains large amounts of them. The aim of our review is to describe the scope of the interaction between MPs and ARGs in the dissemination of environmental antimicrobial resistance and to explore the role of hospital wastewater in aggravating this problem.

## 2. Microplastics: Addressing the Main Problem 

During the last decade, microplastic pollution has been an issue of great concern, representing a growing and emerging threat for human health [6,7]. In 2014, the United Nations Environment Programme (UNEP) identified plastic marine pollution as one of the ten main environmental problems, and in 2015, during the 41st G7 meeting, plastic waste disposal was recognized as a global challenge [8]. Plastic’s huge production and improper disposal are the basis of MPs production [9]. In 2019, the World Health Organization (WHO) showed an alarming presence of MPs in water intended for human consumption, including drinking water, and pointed out the need to implement scientific knowledge on its health implications [10].

The production of plastic materials is relatively recent, since the first manufactured plastic was Parkesine in 1862, but their slow degradation, estimated to go on in the range of hundreds or even thousands of years, represents the main cause of persistence in the environment. 

In 2021, 390.7 million tons of plastic were produced globally with a forecast of 1800 million tons for 2050. In 2021, China reached almost a third of the global plastic production (32%), while Europe showed a more virtuous behavior, dropping from 19% in 2017 to 15% in 2021 of the world production. The 2021 data show that world-leading plastic production sectors are packaging (44%) and building and construction (18%), and the remaining 38% is constituted by automotive, electrical and electronics, household, leisure and sports, agriculture, farming, and gardening [11].

MPs are defined as anthropogenic plastic particles, with sizes between 0.1 µm^−1^ mm (small size) and 1–5 mm (medium size). The particles between 0.001 and 0.1 µm are defined as nanoplastics (NPs). These two categories are defined collectively as micronanoplastics (MNPs). Plastic items whose size is between 5 and 200 mm are called mesoplastics and macroplastics if they are larger than 200 mm.

According to their origin, MPs are divided into two categories. Primary MPs are manufactured by industry and intentionally added to various products to modify their consistency and stability or to confer special features, such as abrasive capacity (medicines and cosmetics, fertilizers and plant protection products, industrial and household detergents, paints and products used in the oil and gas industry); secondary MPs, originated from the fragmentation and breakdown of larger plastic materials, (washing of synthetic garments,17 wear of tires, etc.); or plastic waste released into the environment. The degradation can be physical, chemical, or biological.

MPs are usually composed of polyethylene (total subtypes HD, MD, LD, LLD 29.8%), polypropylene (19.3%), polystyrene (6.7%), polyurethane (7.5%), polyethylene terephthalate (PET) (7.4%), polyvinyl chloride (PVC) (10%), and other compounds (19.3%) [11].

Due to their physical–chemical properties, these materials easily bind and transport chemical contaminants (e.g., antibiotics and heavy metals) and micro-organisms (such as algae and bacteria), increasing their environmental and human health impacts [12,13]. 

In particular, MPs represent a vector for the transport of toxic chemicals in the environment, due to their capacity to adsorb various types of pollutants, to be transported over long distances, and to release these toxic substances into a medium (soil, sediment, or water). MPs from heavily polluted areas have been shown to contain high levels of pollutants. In these areas, a significant number of MPs was demonstrated in the atmospheric fallout. A study conducted in the city of Paris estimated between 2 and 355 particles/m^2^/day of MPs fallout, higher in urban than suburban sites, with 29% made of synthetic fibers. Smaller fibers (200–400 μm and 400–600 μm) were predominant, while larger fibers were rare, with an estimated 3–10 tonnes of fiber/year (2500 km^2^) atmospheric fallout. Although still understudied, the atmospheric compartment has been shown to be a source of MPs [14]. 

Concerning quantities of MPs persist in the environment, especially in marine and aquatic ecosystems, and it is estimated that in the oceans, more than 68% of these are secondary MPs, resulting from the fragmentation of undisposed or incorrectly disposed waste and of their environmental release.

The Mediterranean Sea was found to be particularly polluted by MPs, since it is an enclosed sea surrounded by a heavily populated and industrialized coastline, with many tributary rivers, an important merchant flow, and tourism activity [13,15]. Indeed, recent models predict that the Mediterranean Sea presents one of the highest concentrations of floating plastics in the world: it is estimated that 1,178,000 tons of plastic have accumulated in the Mediterranean Sea, but, so far, most research has been focused on the sea surface, corresponding to less than 0.1% of the total amount [15]. 

To date, 5250 million plastic particles (268,940 tons) are present on the surface of seas and oceans. More than 92% of the plastic particles in the sea are smaller than 5 mm (NP and MP) and represents only 13% by weight [16].

Plastic pollution particles originating from sea- and land-based sources migrate far from the sources, transported by oceanic currents, towards the polar regions, where they remain trapped in aquatic gyres vortices and create accumulation zones of MPs called plastic islands or plastisphere. Oceanic coastlines are less likely to accumulate MPs because of the surface dispersion and transportation of oceanic currents. The largest and most studied plastic island, with the highest concentration of plastic marine debris, spreading over an area about twice the size of France, is called the Great Pacific garbage patch. Located in subtropical waters between California and Hawaii, this ocean region presents a relatively high concentration of ocean plastic, mainly attributed to plastic pollution sources in Asia and intense fishing activity in the Pacific Ocean; its plastic pollution levels are increasing exponentially with a faster rate than in surrounding waters [17]. The adsorption capacity of MPs for organic fouling in the ocean, about six orders of magnitude higher than the surrounding seawater, makes them an ideal substrate for microbial attachment. Microbial colonization and biofilm formation create a new, unique marine ecosystem, evolved to live in man-made plastic environments.

The term plastisphere has been coined to refer to a microbial community of heterotrophs, autotrophs, predators, and symbionts, distinct from the surroundings [7,18,19]. 

Bacterial communities on the plastisphere are selected according to plastic properties and they colonize the MPs surface, produce extracellular molecules, and proliferate. 

Additionally, many of the MPs are aged, so they have undergone a greater degradation due to their permanence in the environment (discoloration, disintegration), and their concentration of pollutants is therefore higher. Mechanical abrasion, solar radiation, and biodegradation cause the depolymerization of MPs and the alteration of their physico-chemical properties, increasing their exchange surface and thus promoting chemical and biological adhesion [16]. Microbial adhesion to MPs determines the formation of biofilm, within which the cited pollutants can accumulate. 

The biofilm formed on MPs surface is considered one of the plausible mechanisms of environmental chemical transport. Examples of toxic substances reported to be adsorbed on MPs include heavy metals (such as Iron, Manganese, Aluminum, Zinc, and Copper) and persistent organic pollutants (such as polyaromatic hydrocarbon and organochlorine pesticide). It has been estimated that the heavy metal concentration reaches up to 38,800 μg/g with a variation between locations, and totals 101,000 ng/g that of organic pollutants [17]. The role of MP biofilm in the environmental transport of toxic chemicals can be deduced from the estimate suggested by Mincer et al. [20], who calculated 1000–15,000 metric tons of microbial biomass associated with floating plastic marine debris. Several factors can influence the adsorption capacity of MPs, such as the type of plastic and their degree of degradation, but this mechanism remains unexplored, mainly because of the lack of a standardized analytical protocol. 

The seafloor is a major sink for MPs. After a variable period, floating plastic can sediment and accumulate at the seafloor, due to biofouling and particle attachment, and sediments constitute a reservoir of persistent contaminants and, over time, durable sources of emission into the water body. Krause demonstrated the seafloor MPs colonization by bacterial communities that create strong local oxygen gradients [21]. 

The accumulation of plastic elements acts as a barrier with a surface redox potential due to the reduction of O_2_ concentrations and the accumulation of reduced compounds such as sulfur and iron. Furthermore, these sediments have a long-term environmental impact because of the sea temperature, which negatively affects the time of microbial degradation. As a result of the formation of these chemical gradients and because of the characteristics of the polymers, the development of specific microbial communities, different from those present in the surrounding environment, is favored.

## 3. Mechanism of Interaction between MPs and ARGs 

The study conducted by a team of Chinese researchers (Nanjing Tech University and Zhejiang University) analyzed polystyrene containers for take-away food and their ability to form MPs and to contribute to increasing antibiotic resistance. This material, once transformed into MPs, can provide an ideal substrate to accommodate not only microorganisms and chemical contaminants but also floating genetic materials containing ARGs. The study described how the aging in the environment of these MPs makes them particularly suitable for binding ARGs-carrying bacteria or directly free-floating genetic materials [22].

The aging process can be caused by mechanical abrasion, solar radiation, and biodegradation, all of which are factors that can increase the exchange surface of MPs, their depolymerization, and, therefore, the alteration of their physical-chemical properties, promoting microbial adhesion and biofilm formation.

It has been shown that exposure to UV-C radiation (254 nm, 0.34 mW/cm^2^), even for only 5–20 days, can increase the size of the pores within the MPs and, therefore, their total volume, thus favoring the binding of ARB, biofilm formation, and ARGs exchange [22].

In biofilm, ARGs can propagate to pathogenic bacteria by horizontal gene transfer, through conjugation (carried by plasmids, iARGs), transformation (free extracellular ARG, eARGs), or transduction (transported by phages, pARGs) [23,24,25].

Although the presence of ARGs in MPs at intracellular and extracellular levels is well demonstrated, their distribution within the biofilm as well as the question of which is the most effective mechanism in their transfer remain unknown [26,27].

The release of chemical compounds has been associated with an alteration in the membrane permeability of microorganisms, which can promote the transfer of ARGs.

Additives or pollutants accumulating on MPs are also influencing factors for the presence of ARGs. Flach et al. have shown that the presence of copper and zinc on plastics, such as those treated with anti-vegetative paint, may favor the binding of ARGs for resistance to macrolides, lincosamides, and aminoglycosides (streptogramin) [28]. Not only inorganic pollutants, such as heavy metals, but also organic ones, such as polycyclic aromatic hydrocarbons, have shown the ability to exert selective pressure on ARGs transfer through co-selection or cross selection. Specifically, it is known that the presence of copper sulfate resistance genes can co-select resistance to different antibiotics, whose genes are located simultaneously on the same genetic element [29].

Among the bacterial species capable of carrying ARGs, *Flavobacterium* spp. and *Chryseobacterium* spp. were the most frequently isolated from MPs. A significant correlation has also been demonstrated between the concentration of MPs and the concentration of integron integrase class-1 gene, known as “mobile genetic element” (MGE), for its ability to promote the transfer of ARGs [30,31].

Non-biodegradable MPs, such as polyethylene terephthalate (PET), have a higher binding capacity to ARGs than biodegradable MPs, such as polyhydroxyalkanoates (PHA).

*Pseudomonas* spp. multi-drug resistant or with glycopeptide resistance genes was found on both PET and PHA, while *Desulfovibrio* spp. resistant to macrolide-lincosamides-streptogramins was mainly isolated on PET [32].

Although non-biodegradable plastic-made MPs are the most studied, the role of biodegradable plastic MPs has not to be undervalued, since their presence is increasing in the environment because of their incomplete biodegradability and their growing use [32,33].

MPs can transport or promote the exchange of ARGs and other pollutants in different environmental compartments, even over long distances [33,34,35,36].

Eckert et al. showed that in the plastisphere, the horizontal gene transfer frequency of ARGs by conjugation is three orders of magnitude higher than in planktonic communities. In the biofilm bonded to MPs, high density and close physical contact between cells facilitates the transfer of ARGs as well as their resistance to external physical agents and, therefore, their environmental persistence [37].

Yang et al. in 2019 performed a metagenomic analysis of the microbiome in one of the largest plastispheres of the globe, the one generated by the North Pacific subtropical vortex, detecting a greater abundance and diversity of ARGs in the extract derived from this matrix compared with the one obtained from the surrounding water [38].

## 4. The Role of Wastewater in Spreading Environmental Antibiotic-Resistance

An important fraction of MPs reaches surface and marine waters after the wastewater treatment, a process able to remove about 90% of the MPs, especially after the tertiary treatment, but still leaving a portion of MPs in the final effluent, which is discharged into water bodies. An important fraction of MPs remains in sewage sludge, which can be used as a compost amendment on agricultural land [39,40].

It is estimated that a medium-sized wastewater treatment facility with a treatment capacity of 5 × 10^7^ m^3^/year can discharge up to 2 × 10^6^ MP/day into the receiving water bodies.

About 67% of the world population has no access to public sewage and, in particular, in low- middle and low-income countries, about 20% of domestic wastewater does not even undergo secondary treatment. In these countries, MPs may be present in higher concentrations in sources of drinking water; however, the health risks associated with exposure to pathogens in untreated or inadequately treated water are considered to be far superior [10].

The selective pressure promoted by the wide and inappropriate use of antibiotics, both in human medicine and veterinary fields, creates strains of ARB in humans and animals that are expelled with urine and feces. This can lead to environmental contamination and favour the transmission of the genetic determinants of antibiotic resistance to other microorganisms living in both the environment and in the microbiome of animals and plants. Furthermore, this situation is aggravated by the co-selection of resistance genes for heavy metals and biocides, which has been reported in wastewater treatment systems. Heavy metals are widely distributed in the environment and can be toxic when they accumulate to a certain concentration. Moreover, especially in the last years, during the COVID-19 pandemic, the use of biocides has increased within domestic, industry, and healthcare settings for disinfection purposes. Exposure to these non-antibiotic antimicrobial agents can induce or select for bacterial adaptations, which results in decreased susceptibility to one or more antibiotics. Co-selection of resistance to antibiotics, biocides, and heavy metals represents a further challenge to be addressed [41,42].

Wastewater treatment plants, collecting wastewater from a wide variety of sources, are considered hotspots for the diffusion of ARGs as they are sites of potential bacterial growth and genetic exchange [3,43]. Activated sludge purification provides a favorable environment for the development of active biomass (saprophytic bacteria, protozoa, amoeba, rotifers, and other microorganisms), the basis of biological purification systems, but also a matrix for genetic exchange between bacterial species. In fact, sewage treatment plants are potential antibiotic resistance incubators.

Concentrations of antibiotics after treatment increase significantly compared to their upstream concentrations by an average factor of eight times, except clindamycin. These concentrations remain high even in the effluent at about 3 km. On the contrary, removal of ARGs is more effective. The persistence of antibiotics in these waters can increase the potential selection of ARGs and promote transfer between bacterial populations [44].

Landfills, receiving the sewage sludge, provide a favorable environment for the proliferation of ARB, which can transfer ARGs, by conjugation, to other environmental bacterial strains, thereby contributing to the spread of antibiotic-resistance and adverse public health effects. Landfill leachate is a dangerous liquid, containing antibiotics and toxic organic pollutants, which can promote the selection of ARB and ARGs in the environment [45].

In this context, hospital wastewaters are considered hotspots for antibiotic resistance. Compared to urban wastewaters, the hospital ones contain a variety of toxic or persistent substances, such as pharmaceuticals, radionuclides, disinfectants for medical purposes in a wide range of concentrations, and, in particular, pathogen microorganisms containing ARGs [46,47]. Although there is still a lack of data in literature focused on the presence of MPs in hospital wastewaters, the role of these effluents in the diffusion of ARGs is well documented. The most recent and representative articles on this topic are summarized in Table 1. 

Several studies demonstrate that the main source of ARGs is represented by hospital wastewater [48,64], which is released in domestic sewage treatment plants and, after appropriate treatment, discharged into water bodies. It shall also be considered that other sources are represented by waste and runoff-water from zootechnical plants. Depuration plants play a fundamental role in decreasing the environmental burden of antibiotics and ARG, although the treatment can only partially metabolise and/or remove them, with varying degrees of efficacy [64,65,66], thus contributing to the resistance strains selection. A study focused on the release of antibiotics and ARGs in wastewater highlighted the highest concentration of fluoroquinolones in hospital effluent samples. The abundance of this antibiotic family may be linked to its frequent use in hospital practice [66].

Several reports have already described in Italy the presence in wastewater of resistance determinants [67]. Despite the diffusion in hospital settings of resistant bacteria, this is mostly due to the circulation of particular “high risk” clones, and the persistence of ARGs in environment isolated strains can also be attributed to their presence on mobile genetic elements (Plasmids and Transposons). Horizontal and inter/intra- species transfer of ARGs, also through co-selection processes, can be facilitated by the acquisition of genes that confer resistance to heavy metals and biocides––chemical stressors that are regularly found in treated water [68].

Italy is considered endemic for the diffusion of Metallo-β-lactamases (MBLs) producing Enterobacteriaceae, which can confer resistance to most β-lactams, such as cephalosporins and carbapenem, used for treating the most severe infections; despite the decreasing trend in prevalence of carbapenemase-producing *Klebsiella pneumoniae* from 2015 to 2019, an increase in *E. coli* was noticed. The resistance to carbapenems has also spread between the *Acinetobacter* species, often combined with fluoroquinolones and aminoglycosides-resistance. Among Gram+, the rise of vancomycin-resistant *Enterococcus faecium* and Methicillin-resistant *Staphylococcus aureus* was documented [2].

As reported by Verlicchi et al. [69], concentrations of micropollutants (e.g., antibiotics, analgesics, and heavy metals) in hospital wastewaters (HWW) are between 4 and 150 times higher than in urban wastewaters.

ARB and ARGs persist in depuration plant effluents, also after the final disinfection, flowing into surface water. Their presence in supply sources can constitute a relevant health risk. In this context, the role of MPs is emerging, whose persistence and accumulation in the environment contribute to the diffusion of antimicrobial resistance, even at long distances. 

The depolymerisation of MP, due to their degradation in the environment, promotes microbial adhesion and biofilm formation, providing an ideal substrate not only for microorganisms and chemical contaminants but also for free-floating genetic material containing ARGs. 

Their aging increases the chance for ARGs bonding because of their depolymerization and the alteration of their chemical-physical properties, thus favoring microbial adhesion and biofilm formation. 

Inside the biofilm, ARGs can spread to pathogenic bacteria by horizontal gene transfer, transformation, and transduction [23,24,25,35,36].

In this scenario, the role of wastewater depuration is still poorly documented, especially referring to its ability to partially remove MPs, particularly after the tertiary treatment, albeit leaving a certain amount of them in the final effluent [39,40]. 

Considering the aggravation of this problem, new studies have been set up, although they are still limited and often have discordant conclusions. The methods to reduce the load of ARB, their vectors, and drugs in wastewater can enable the improvement of disinfection and wastewater treatment processes [3,68]. 

In Italy, in accordance with the main areas of intervention identified in the National Action Plan on Antimicrobial Resistance (PNCAR 2017–2020), one of the priority actions is the use of innovative methods in the depuration treatment of urban and hospital wastewater in order to intercept the greatest number of contaminants before they can reach environmental matrices. This action is also a goal of the National Recovery and Resilience Plan (PNRR) that provided for specific investments aimed at strengthening the management and protection of water and marine resources. According to the Sustainable Development Goals (SDGs), adopted by the United Nations in 2015, safe water is not only necessary for drinking and sanitation (SDG 3: Good health and well-being, SDG 6: Clean water and sanitation) but also for irrigation of crops and, thus, for nutrition (SDG 2: Zero hunger), for food production economy (SDG1: No poverty), and (by discharging effluent) for ecosystems and oceans (SDG 14: Life below Water). 

## 5. Potential Impact on Health of MPs and Their Contaminants

Because of their widespread dissemination, persistence, and impact on the environment, MPs, ARB, and ARGs are the cause of rising concern about their potential effect (direct and indirect) on human health [70,71].

To date, several pieces of evidence demonstrate that MPs can access the human body through ingestion (i.e., seafood) and inhalation; following their entrance in pulmonary and gastrointestinal systems, their accumulation and distribution in human tissues are directly related to the particle size [72].

The mainly explored pathway of human exposure to MPs through ingestion is represented by their accumulation in aquatic organisms. The continuous release of non-degradable MPs in aquatic environments contributes to the permanent pollution of these ecosystems and, consequently, to the entrance of MPs into the food chain. As reported in the literature, in the case of environmentally relevant concentrations of MPs (≤1 mg/L), aquatic organisms can accumulate high quantities in their digestive and respiratory systems, leading to early death [73]. The genotoxicity of realistic MP concentrations to aquatic organisms was meta-analyzed by Sun et al. in 2021, demonstrating that the genotoxic damage in the MPs-treated group increased by 24% compared to the control [74]. This relevant impact has also been proved by a study conducted along the Mediterranean coast of Turkey, where 70% of the 1822 microparticles extracted from the digestive system of 1337 fish samples were MPs [75]. 

Another vehicle for MPs into the human body is represented by tap water and bottled water. It is estimated that 80% of urban tap water worldwide is contaminated with MPs with a concentration ranging from 1383 to 4464 particles/L in raw water and from 243 to 684 particles/L in treated water [76].

Regarding bottled water, some studies showed that the most frequently detected synthetic polymer residues are usually the main components of the bottles themselves, identifying the packaging to be among the main sources of contamination [77,78,79].

For these reasons, the European Food Safety Authority (EFSA) points to a potential risk linked to the presence in food and water of high concentrations of pollutants, such as polychlorinated biphenyls (PCBs) and polycyclic aromatic hydrocarbons (PAHs), in addition to the presence of residues of compounds used in packaging, such as bisphenol A (BPA), a genotoxic compound and endocrine disruptor [80].

It is assumed that the uptake of microparticles by M cells in Peyer’s patches in the ileum is the main route of their entry into the circulatory system. Through this path, they are able to reach the intestine lymphatic system and then the liver before being excreted [81].

The distribution and accumulation of MPs have been demonstrated in mice tissue of the liver, kidney (nephrotoxicity), and intestines (gastrointestinal toxicity) [82].

Experiments in rodents show that lymph nodes can adsorb up to 0.3% of plastic fragments smaller than 150 μm, while MPs smaller than 110 μm have been detected in the portal vein and even in organs when MPs are smaller than 20 μm in size [73].

Furthermore, another route of entrance is represented by the respiratory system. MPs from atmospheric fallout can enter the body mainly through inhalation, causing toxic reactions and disturbances in various organs and systems and even exposing animals and humans to a potential cancer risk. Smaller MPs are not removed from muco-ciliary clearance and reach the alveoli, where these fibers might lead to lung inflammation and potentially secondary genotoxicity [83]. A recent in vivo work demonstrated that inhalation of MNPs (diameter of 5 μm) induces pulmonary fibrosis in a dose-dependent manner in mice, activating an intense oxidative stress in the lungs [84].

The impact of MPs on the cardiovascular system was identified by Li et al. After exposing rats to 0.5 μm MNPs at 0.5, 5 and 50 mg/L for 90 days, they found increased Troponin I and creatine kinase-MB (CK-MB) levels in serum, structural damage, and apoptosis of myocardium cells and collagen proliferation [85].

The MPs food-borne exposure associated risk for human health comes from the particles themselves, which present a physical hazard (related to their dimension), a chemical hazard (due to unbound monomers, additives, and sorbed chemicals from the environment), and a biological hazard from microorganisms that may attach and colonize on MPs (biofilms).

Jin et al. in 2018 detected inflammation and dysbiosis of the intestinal microbiota in the adult Zebra Fish after ingestion of polyethylene MPs; Kesy et al. in 2017 detected the presence of *Mytilus Edulis* intestinal microbiome on the particles of MP expelled into the environment after passing through its intestine, as well as a rapid colonization of these MPs by pathogenic bacteria in the environment [86,87].

An important site of accumulation of MPs is the human gut, where they can lead to increased or decreased alteration in the amount of mucus, changes in gut microbiota composition, intestinal cell inflammation, loss of tight junction protein, and recall of immune cells [88,89].

MPs, enriched in microorganisms and ARGs, might affect the intestinal flora changing the specific conditions of the human intestinal tract and carrying potential pathogens. These mechanisms could contribute to the growth of antibiotic resistance in humans [90].

The spread of ARGs and MNPs in soil environments facilitates the migration of ARGs into the food chain, raising concerns about their potential risk to human health through food consumption [91].

While there is much knowledge of the direct effects of microplastics on human health, we still know little about the indirect effects of plastic pollution, including their role in the transport of antibiotic resistance. For this reason, further studies on the interactions between ARGs/MNPs and environmental matrices would be needed, as well as the dual toxicity of ARGs and MNPs with other coexisting chemical substances and their trophic transfer would deserve more attention [92,93]. 

## 6. Conclusions

The environmental ubiquity of MPs is a known threat to human health. In fact, MPs’ presence was confirmed in supply sources and even in food and drinking water. 

This problem is aggravated by the capacity of MPs to act as a “Trojan horse”, adsorbing chemicals and micro-organisms from the surrounding environment, through biofilm formation. 

It has been estimated that the presence of MPs in the effluents of sewage treatment plants promote the persistence of these organic and inorganic contaminants, adsorbed as part of biofilm on MPs surface, even after final disinfection, and, consequently, they can be detected in surface watercourses. 

For this reason, MPs are considered a vector of antibiotic resistance diffusion at long distance, too. This phenomenon is particularly important for hospital wastewater, where the presence of ARGs and antibiotics is greater compared to urban wastewater, enabling the selection and horizontal transfer of ARGs. In conclusion, the presence of ARGs on MPs is an emergency that shall be managed in the near future through a correct management of the disposal of plastic waste and an adequate treatment of wastewater, particularly hospital effluent, as a part of the core-component against antibiotic resistance diffusion.

With this aim in mind, it is essential to deepen the knowledge on the environmental behavior of MPs and their role in the transmission of ARGs, on the study of MP’s resistome and its transfer potential to the surrounding environment, to better understand the risk and extent of exposure for humans and the effects on health.

## Figures and Tables

**Table 1 ijerph-20-05868-t001:** ARGs and antibiotics concentrations in untreated hospital wastewater.

Author	Location	Sampling Years	Hospital Capacity	Average ARGs Absolute Abundance, Minimum and Maximum Value (Gene Copies/L)	Antibiotics Abundance Average, Minimum and Maximum Value (µg/L)
Rodriguez-Mozaz et al., 2015 [48]	Girona, Spain	2011–2012	Main hospital, 400 beds	*bla*_TEM_: ~10^7^; *qnr*S: ~10^7^; *erm*B: ~10^9^; *sul*I: ~10^8^; *tet*W: ~10^9^	CIP: 8.305–13.779; OFX: 4.750–14.378; CFZ: 0.045–0.083; CTX: 0.144–0.240; AZM: 0.020–0.059; CLAR: 0.167–0.941; SMX: 4.817–0.190; TMP: 3.787–0.136; MTR: 1.793–0.524
Hutinel et al., 2022 [49]	Gothenburg, Sweden	2015–2019	Sahlgrenska University Hospital, 2000 beds	* *mcr-*1: 9.96 × 10^−7^–2.25 × 10^−5^; *mcr-*3: 4.34 × 10^−5^–1.45 × 10^−3^; *mcr-*4: 4.88 × 10^−6^–3.20 × 10^−5^; *mcr-*5: 4.07 × 10^−5^–1.19 × 10^−4^; *sul*4: 7.53 × 10^−6^–7.14 × 10^−5^; *gar*: 8.53 × 10^−7^–5.09 × 10^−4^; *optr*A: 2.60 × 10^−5^–4.74 × 10^−5^; *cfr*(A): 3.20 × 10^−7^–7.61 × 10^−6^	Data not available
Gönder et al., 2021 [50]	Istanbul, Turkey	2 years	Medical faculty hospital, 1358 beds	Data not available	AMP: 0.41; AZM: 1.06; CIP: 3.48 CLAR: 5.34; CLM: 0.1; ERY: 0.28; MTR: 0.86; NOR: 0.1; OFX: 0.96; SDZ: 0.24; SMX: 15.68; SPR: 1.93; TMP: 3.6
Yao et al., 2021 [51]	East China	2019	Primary Hospital, 80 beds Secondary Hospital, 150 beds Tertiary Hospital, 964 beds	*qeq*A: 5.73 × 10^5^; *qnr*A: 1.44 × 10^8^; *qnr*D: 2.02 × 10^7^; *qnr*S: 1.82 × 10^8^; *bla*_OXA-1_: 2.22 × 10^10^; *bla*_TEM-1_: 7.17 × 10^9^; *bla*_GES-1_: 1.33 × 10^10^; *bla*_OXA-10_: 3.50 × 10^9^; *bla*_SHV-1_: 1.16 × 10^7^; *bla*_DHA-1_: 2.02 × 10^8^	CN: 0.03–0.88; CEF: 0.02–0.11; CED: 0.37–2.38; CAZ: 0.14–31.21; AMX: 0.04–1.43; AMP: 0.14–0.67; OFX: 1.39–49.47; NOR: 0.05–0.61; TMP: 0.02–0.50; FOX: 0.36–8.96; CFZ: 0.45–5.01; CEP: 106.76–540.39; MER: 0.02–0.2
Wang et al., 2018 [52]	XinXiang city, Central China	2016	Tertiary Hospital 1, 1740 beds Tertiary Hospital 2, 1200 beds Tertiary Hospital 3, 700 beds	*tet*X: 1.30 × 10^9^–4.14 × 109; *tet*M: 4.10 × 10^7^–4.15 × 10^8^; *tet*O: 1.38 × 10^9^–8.34 × 10^10^; *ere*A: 6.81 × 10^7^–3.50 × 10^8^; *erm*A: 1.88; *erm*B: 8.28 × 10^8^–3.20 × 10^9^; *sul*1: 1.30 × 10^10^–7.24 × 10^10^; *sul*2: 1.29 × 10^9^–7.83 × 10^9^; *sul*3: 6.40 × 10^5^–4.84 × 10^7^; *qnr*A: 1.01× 10^7^–4.79 × 107; *qnr*D: 1.73 × 10^6^–3.72 × 106; *oqx*B: 3.34 × 10^6^–1.85 × 10^7^	SDZ: 0.123; SMX: 0.533; OFX: 2.329; NOR:1.629; CIP: 1.334; TET: 1.411; OTC: 1.479; ERY: 0.479; LN: 0.132; TMP: 0.268; CN: 2.392
Yilmaz et al., 2017 [53]	Istanbul, Turkey	2014	Medical Faculty Hospital 1, 1358 beds Medical Faculty Hospital 2, 1285 beds Training and Research Hospital, 612 beds	Data not available	SMX: 0.55–8.5; CLN: <0.01–4.1; TMP: <0.01–2.2; CIP: 1.9–24; CAZ: 1.6; AZM: <0.01–0.4; CLAR: 0.063–15; SDZ: <0.01–0.16; OFX: 0.082–200; CLM: <0.01–0.036; MTR: <0.03–3; SPR: <0.01–0.047
Azanu et al., 2018 [54]	Kumasi, Ghana	2014	Komfo Anokye Teaching Hospital, 1200 beds University Hospital, 120 beds	Data not available	CIP: 0.247–0.420; ERY: 11.352–15.733; TMP: 7.944–10.613; SMX: 0.094–4.826; AMX: 0.058–0.116; AMP: 0.075–0.252; CFX: 0.016–0.024; MTR: 0.024–0.120; DOX: 0.002–0.006; TET: 0.107–0.324; CTC: 1.052–1.557; OTC: 2.315–3.590
Zhu et al., 2021 [55]	Eastern China	2018–2019	Three public hospitals	*bla*_NDM1–15_: 3.13 × 10^7^–1.72 × 1010; *bla*NDM16: 3.13 × 10^7^–1.72 × 1010; *mcr*-1–*mcr*-5: 2.03 × 10^6^–1.32 × 10^9^; *tet*(X1)–*tet*(X5): 5.62 × 10^5^–3.69 × 10^9^	Data not available
Sib et al., 2020 [56]	Germany	2016–2018	Maximum care hospital, 1274 beds	*bla*_NDM_: 10^8^–10^9^; *bla*_VIM_: 10^8^–10^9^; *bla*_CTX-15-M_: ~10^7^; *sul*1: 10^10^–10^11^; *mcr*-1: ~10^3^–10^5^	Data not available
Kayali and Icgen, 2020 [57]	Ankara, Turkey	2017	Six major hospitals H1, 160 beds H2, 270 beds H3, 468 beds H4, 484 beds H5, 730 beds H6, 1140 beds	*aadA*: 9.5 × 10^7^; *tet*A: 2.5 × 10^7^; *cmlA*: 8.3 × 10^6^; *sul*1: 3.7 × 10^6^; *qnr*S: 1.2 × 10^6^; *erm*B: 2.9 × 10^5^; *bla*_CTX-M_: 1.1 × 10^3^	Data not available
Wang et al., 2022 [58]	Jeddah, Saudi Arabia	2020	Infection Disease Hospital (Hospital A) Psychiatric hospital (Hospital B)	Data not available	SMX: 0.013–0.330; ERY: 0.00006–0.006; CIP: 0.015–0.068; MFX: 0.008–0.009; PG: 0.008–0.017; MER: 0.013; TET: 0.005–0.006
Kosma et al., 2020 [59]	Ioannina, Greece	1 year	Ioannina University Hospital: 800 beds	Data not available	SDZ: 0.348; SMX: 0.525; SPR: 0.262; STZ: 0.109; TMP: 0.295
Vazquez et al., 2023 [60]	Galicia, Spain Porto, Portugal	Data not available	Hospital 1 Hospital 2	Data not available	AZM: 1.9–3.4; CIP: 0.18–0.66; OFX: 2.5–3.5; SDZ: 0.35; SMX: 0.073–0.10; TMP: 0.13–0.20; CLAR: 0.034; ERY: 0.063; NOR: 0.21
Dinh et al., 2017 [61]	Fontenay-les-Briis, France	2009–2014	Polyvalent medical centre: 360 beds	Data not available	ERY: 0.80; TET: 0.03; AMX: 0.11; TMP: 0.94; ORM: 0.04; SMX: 2.1; PI: 0.02; ENO: 0.76; LOM: 0.25; NOR: 12.1; CIP: 5.8; OFX: 17.9; VAN: 3.6
Le et al., 2016 [62]	Singapore	2014	2 Hospital Blocks, 1597 beds Block A, clinical isolation wards Block B, general wards Hospital 2, 1500 beds	*bla*_NDM_: 2.29 × 10^9^; *bla*_KPC_: 4.08 × 10^10^; *bla*_CTX-M_: 1.25 × 10^9^; *bla*_SHV_: 6.19 × 10^8^	AZM: 0.11–1.51; CIP: 1.72–76.44 CHL: 0.65; CLAR: 0.76–72.87; ERY: 0.3–17.63; MER: 0.19–1.01; SMX: 0.94–28.36; TMP: 0,78–71.8; CLM: 1.43; LN: 0.015; TET: 0.82
Szekeres et al., 2017 [63]	Cluj County, Romania	2015	Oncological Hospital, 535 beds General Hospital, 113 beds General Hospital, 453 beds	*aac*C2: 4.41 × 10^8^–1.08 × 10^9^; *bla*_VIM_: 1.43 × 10^5^–3.19 × 10^6^; *cat*A1: 5.31 × 10^6^–1.44 × 10^8^; *erm*A: 2.16 × 10^5^; *flo*R: 1.58 × 10^4^–1.73 × 10^8^; *tnp*A: 1.41 × 10^9^–2.76 × 10^9^; *mef*A: 8.01 × 10^7^–1.91 × 10^8^; *qac*EΔ1: 2.22 × 10^9^–3.80 × 10^6^; *bla*_SHV_: 1.94 × 10^8^–3.41 × 10^5^; *sul*I: 6.11 × 10^9^–1.51 × 10^10^; *sul*II: 4.29 × 10^7^–5.75 × 10^8^; *tet*A: 2.69 × 10^8^–5.94 × 10^8^; *tet*B: 1.13 × 106–1.66 × 10^6^; *tet*C: 1.33 × 10^5^–1.92 × 10^5^; *tet*O: 4.91 × 10^7^–1.09 × 10^9^; *tet*W: 6.79 × 10^6^–1.69 × 10^8^	AMP: 8.07–53.05; CAZ: 3.66–10.46; CEP: 5.18–8.52; TMP: 13.06–30.38; IPM: 14.42; TAZ: 10.26; VAN: 5.03–13.98; PIP: 7.81; ERY: 7.52; SMX: 6.06; TET: 1.34; GM: 7.87

**Abbreviations:** AMX: amoxicillin; AMP: ampicillin; AZM: azithromycin; CED: cefradine; CN: cefalexin; CEF: cefalotin; CFZ: cefazolin; CEP: cefepime; CTX: cefotaxime; CAZ: ceftazidime; CHL: chloramphenicol; CTC: chlorotetracycline; CLN: cilastatin; CIP: ciprofloxacin; CLAR: clarithromycin; CLM: clindamycin; DOX: doxycycline; ENO: enoxacin; ERY: erythromycin; FOX: cefoxitin; GM: gentamicin; IPM: imipenem; LN: lincomycin; LOM: lomefloxacin; MER: meropenem; MFX: moxifloxacin; MTR: metronidazole; NOR: norfloxacin; OFX: ofloxacin; ORM: ormethoprim; OTC: oxytetracycline; PG: penicillin G; PI: pipemidic acid; PIP: piperacillin; SDZ: sulfadiazine; SMX: sulfamethoxazole; STZ: sulfathiazole; SPR: sulfapyridine; TAZ: tazobactam; TET: tetracycline; TMP: trimethoprim; VAN: vancomycin. * ARGs relative abundance in copies per 16S rRNA gene copies.

## Data Availability

No new data were created or analyzed in this study. Data sharing is not applicable to this article.

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
