# Peer review of "Microplastics and Antibiotic Resistance: The Magnitude of the Problem and the Emerging Role of Hospital Wastewater"

_ijerph, 2023, doi:10.3390/ijerph20105868_

Round 1
Reviewer 1 Report
The article “Microplastics and antibiotic resistance: the role of hospital wastewater” is devoted to very important problem the role of microplastics in the spread of antibiotic resistance genes. The authors made a good review on the problem of environmental pollution by microplastics and their role in the spread of antibiotic resistance genes.
While reading the article, I’ve got some comments:
L.82 “…microbial agents (such as algae and bacteria)..” Algae cannot be microbial agents
L.339. “It is estimated that 80% of urban tap water worldwide is contaminated with MPs, though the levels of MPs detected in treated water are lower (a maximum of 684 particles/L) compared to raw samples (a maximum of 4464 particles/L)”.
To my mind the level of MPs in tap water is to much! Сheck this data, especially since I did not find such data in the link [62]
After correcting the text, taking into account the comments, the article can be published.
Author Response
Dear Reviewer,
Thank you very much for the attention paid to our paper. Considering your comments, we modified the manuscript accordingly with the suggestions and we reported the corrections in red in the text.
With kind regards
Beatrice Casini
Reviewer 1
R: L.82 “…microbial agents (such as algae and bacteria).” Algae cannot be microbial agents.
A: Thanks for your suggestions and comments. We took into account your corrections as follows:
L82: we changed the words "microbial agents" with "micro-organisms" as you suggested.
R: L.339. “It is estimated that 80% of urban tap water worldwide is contaminated with MPs, though the levels of MPs detected in treated water are lower (a maximum of 684 particles/L) compared to raw samples (a maximum of 4464 particles/L)”. To my mind the level of MPs in tap water is too much! Сheck this data, especially since I did not find such data in the link [62]
A: 339: thank you for pointing it out. We realised that the reference was not accurate. We corrected it with the following reference: “Eerkes-Medrano D, Leslie HA, Quinn B, Microplastics in drinking water: A review and assessment, Current Opinion in Environmental Science & Health,Volume 7,2019, Pages 69-75, ISSN 2468-5844. doi: 10.1016/j.coesh.2018.12.001.” We also modified the text reporting the range of particles/L found in cited article.
Reviewer 2 Report
The role of microplastics (MPs) in the transmission of antibiotic resistance genes (ARGs) has attracted increasing research attention worldwide due to its unique ecological and environmental effects. The author first introduces some basic concepts of microplastics and resistance genes, and then introduces the status quo of antibiotics in wastewater.
1. The author should reconsider the framework of the article and its title, which is a bit grand. It's easy for readers to get confused about the key point or innovation of this article.
(eg. 2. MPs and Antibiotic-Resistance;3. The role of wastewater; 4. Health effects)
A subtitle is a line of text below the main title. It is a further explanation of the main title and is used to help readers understand the specific content of the article.
2. The status quo or transmission path of resistance genes of microplastics, or the status quo of resistance genes/antibiotics in wastewater can be considered to be presented in charts or tables. There is no chart or table in this paper, which does not quite meet the requirements of the review.
3. The author should further summarize and conclude the draft. There are a large number of 1-2 sentences or 2-3 sentence paragraphs in this draft, and the writing logic needs to be improved.
The quality of English language
Author Response
Dear Reviewer,
Thank you very much for the attention paid to our paper. Considering your comments, we modified the manuscript accordingly with the suggestions and we reported the corrections in red in the text.
With kind regards
Beatrice Casini
Reviewer 2
R: The author should reconsider the framework of the article and its title, which is a bit grand. It's easy for readers to get confused about the key point or innovation of this article(eg. 2. MPs and Antibiotic-Resistance;3. The role of wastewater; 4. Health effects) A subtitle is a line of text below the main title. It is a further explanation of the main title and is used to help readers understand the specific content of the article.
A: Thank you for the suggestion. We modified the main title as reported “Microplastics and antibiotic resistance: the magnitude of the problem and the emerging role of hospital wastewater” and paragraphs’ title as below: 1. Introduction; 2. Microplastics: addressing the main problem; 3. Mechanism of interaction between MPs and ARGsï¼›4. The role of wastewater in spreading environmental antibiotic-resistanceï¼› 5. Potential impact on health of MPs and their contaminants; 6. Conclusion
R: The status quo or transmission path of resistance genes of microplastics, or the status quo of resistance genes/antibiotics in wastewater can be considered to be presented in charts or tables. There is no chart or table in this paper, which does not quite meet the requirements of the review.
A: Thank you for pointing it out. We added a table showing the most representative and recent articles worldwide about the presence of ARGs and antibiotics in hospital wastewater. We reported it in the text.
R: The author should further summarize and conclude the draft. There are a large number of 1-2 sentences or 2-3 sentence paragraphs in this draft, and the writing logic needs to be improved.
A: Thank you for your comment. We extended and rephrased the conclusions, and we marked it in red in the text.
Reviewer 3 Report
The paper can be accepted after revision and taking into account the following remarks:
Introduction
1. Nothing is written about MP in hospital wastewater while it is included in the title
2. No goal of the study
Other sections
According to the title, main focus of this study must be on the role of hospital wastewater on MP-antibiotic relations. In fact, hospital wastewater is discussed only in small part of the manuscript (a bit more than 1 page). I recommend to extend the analysis of the role of hospital wastewater.
Conclusion
I did not find any estimation of influence of adsorption by MP in the paper although this is written in the conclusion.
Author Response
Dear Reviewer,
Thank you very much for the attention paid to our paper. Considering your comments, we modified the manuscript accordingly with the suggestions and we reported the corrections in red in the text.
With kind regards
Beatrice Casini
Reviewer 3
R: Introduction
- Nothing is written about MP in hospital wastewater while it is included in the title
- No goal of the study
A: Thank you for all your suggestions and comments. We modified the introduction as you suggested and we added a short introduction about hospital wastewater, also specifying the aim of the review. Modifications are reported in red in the text.
R: Other sections
According to the title, main focus of this study must be on the role of hospital wastewater on MP-antibiotic relations. In fact, hospital wastewater is discussed only in a small part of the manuscript (a bit more than 1 page). I recommend extending the analysis of the role of hospital wastewater.
A: Thank you for pointing it out. We modified the title, and we extended the discussion about the role of hospital wastewater throughout the text adding a summary table with the most representative and recent articles worldwide about the presence of ARGs and antibiotics in hospital wastewater.
R: Conclusion
I did not find any estimation of influence of adsorption by MP in the paper although this is written in the conclusion.
A: Thank you for the suggestion. We extended the description of MPs adsorption as follows, and we added it in paragraph 2 (Microplastics: addressing the main problem).
“In particular, MPs represent a vector for the transport of toxic chemicals in the environment, due to their capacity to absorb various types of pollutants, to be transported over long distances and to release these toxic substances into a medium (soil, sediment or water. (…) The biofilm formed on MPs surface is considered one of the plausible mechanisms of environmental chemicals transport. Examples of toxic substances reported to be adsorbed on MPs include heavy metals (such as Iron, Manganese, Aluminium, Zinc, Copper) and persistent organic pollutants (such as polyaromatic hydrocarbon and organochlorine pesticide). It has been estimated that the heavy metal concentration reaches up to 38,800 μg/g with a variation among locations and 101,000 ng/g for organic pollutants [17] .The role of MPs biofilm in the environmental transport of toxic chemicals can be deduced from the estimate suggested by Mincer et al. [20], who calculated 1000–15000 metric tons of microbial biomass associated with floating plastic marine debris. Several factors can influence the adsorption capacity of MPs, such as the type of plastic and their degree of degradation, but this mechanism remains still unexplored, mainly because of the lack of a standardized analytical protocol. “
Round 2
Reviewer 2 Report
The author has made a detailed revision according to the expert's opinion, which has met the requirements for publication.